# Acupuncture (Jin's Three-Needle) versus sham acupuncture in treating mild-to-moderate depression: Study protocol for randomized clinical trial

Hao Wen[1,2], Jiafeng Wang[1,2], Kaiying Zhang[1], Na Zhang[1], Yiqing Shan[1], Xiaolan Yang[1], Pengcheng Zhang[1], Xingyu Xiang[3], Baoguo Zhang[1]*, Chong Zhang[4]*

1 Guangzhou Huangpu District Hospital of Traditional Chinese Medicine, Guangzhou, China, 2 School of Traditional Chinese Medicine, Jinan University, Guangzhou, China, 3 The Affiliated TCM Hospital of Guangzhou Medical University, Guangzhou, China, 4 First Affiliated Hospital of Guangzhou University of Chinese Medicine, Guangzhou, China

* 13737940868@163.com (CZ), zbg57289@163.com (BZ)

## Abstract

### Background

Mild-to-moderate depression is an increasing public health problem worldwide, which is more prevalent than major depression but has received less attention Currently, no existing highly effective pharmacological treatments with minimal adverse reactions are available for this condition. Jin's Three-Needle Acupuncture (JTN) may be an alternative treatment option for mild-to-moderate depression patients.

### Methods

This is a prospective, parallel-arm, single-blind, randomized controlled trial. We plan to enroll 106 participants with mild-to-moderate depression, randomly assigning them to receive either JTN or sham acupuncture (SA) in a 1:1 ratio. All participants will receive the JTN treatment or SA treatment for 8 weeks (3 per week). The primary outcome will be assessed using the 17-item Hamilton Depression Rating Scale (HAMD-17). Secondary outcomes will include the Self-Rating Depression Scale (SDS), Pittsburgh Sleep Quality Index (PSQI), Hamilton Anxiety Scale (HAMA), Self-Rating Anxiety Scale (SAS), and the Traditional Chinese Medicine Syndrome Score Scale (TCMSS). Evaluations will be performed at baseline and at weeks 4, 6, and 8 after treatment initiation. Statistical analyses will be conducted on both the intention-to-treat (ITT) and per-protocol (PP) datasets.

### Discussion

This study aims to provide high-quality evidence regarding the efficacy and safety of JTN as a treatment for mild-to-moderate depression. Additionally, the mechanisms underlying the effects of JTN in treating mild-to-moderate depression will be explored.

**Data availability statement:** No datasets were generated or analysed during the current study. All relevant data from this study will be made available upon study completion.

**Funding:** This work is funded by the Guangzhou Region Major Science and Technology Project on Traditional Chinese Medicine(2025QN018). The funder has no role in study design, data collection, analysis and interpretation, decision to publish, or preparation of the manuscript.

**Competing interests:** The authors have declared that no competing interests exist.

**Abbreviations:** JTN, Jin's three-needle acupuncture; TCM, Traditional Chinese medicine; HAMD-17, The 17-item Hamilton Depression Rating Scale; SDS, Self-rating depression scale; PSQI, Pittsburgh sleep quality index; HAMA, Hamilton Anxiety Scale; SAS, Self-Rating Anxiety Scale; TCMSS, the Traditional Chinese Medicine Syndrome Score Scale; ITT, The intention-to-treat; PP, per-protocol datasets; AEs, Adverse events; CRFs, Case report forms;

## Trial registration

The trial registered with the International Traditional Medicine Clinical Trial Registry (ITMCTR2024000872). URL: https://www.itmctr.ccebtcm.org.cn

## Introduction

Depression, a prevalent psychiatric disorder characterized by persistent low mood, represents a significant contributor to the global disease burden and a leading cause of disability [1]. In China, previous studies [2] have shown that the annual prevalence rate of depression in China was merely 2.06% between 2001 and 2005, but this figure rose to 10.6% by 2022 [3]. A particular difficulty in treating depression lies in the large number of people experiencing mild to moderate symptoms. Despite making up a substantial portion of those affected, these individuals often fall through the cracks of clinical care. Without proper intervention at this stage, their condition may worsen into more severe forms of depression [4]. Consequently, investigating potential improvements in the treatment of mild-to-moderate depression is of critical importance.

Current treatment guidelines present clinicians with something of a dilemma. First-line antidepressant pharmacotherapy, while effective for many, is often associated with a range of adverse effects that can impact cardiovascular, gastrointestinal, and endocrine systems, raising concerns about their suitability for milder clinical presentations [5–7]. International guidelines indeed advise caution in the routine prescription of antidepressants for mild-to-moderate cases, often recommending initial monitoring or psychotherapy [8,9]. However, psychological therapies such as cognitive behavioural therapy (CBT) are highly dependent on the availability of trained psychotherapists, making them difficult to implement in low-income areas [10]. Therefore, it is crucial to seek a potential treatment that is safe, accessible, affordable, and effective.

Jin's Three-Needle (JTN), a distinct form of acupuncture, has been clinically applied for decades and is recognized as a unique school of acupuncture in Southern China. JTN demonstrates therapeutic efficacy in treating mental disorders and assisting patients in modulating their mental states [11]. Our earlier research shows that JTN treatment may considerably reduce the major symptoms of depression for opioid dependence patients [12]. Recent studies have demonstrated that peripheral blood leukocytes from patients with depression exhibit elevated expression levels of depression-related genes, particularly inflammatory markers, which often display DNA methylation patterns resembling those observed in brain tissues [13,14]. For instance, methylation of the glucocorticoid receptor gene NR3C1 and the serotonin transporter gene SLC6A4 in leukocytes has been linked to depressive symptoms [15,16]. DNA methylation in peripheral blood leukocytes represents a promising biomarker for depression, as it frequently suppresses or blocks the transcription of genes associated

with depressive symptoms [17,18]. Research has indicated that acupuncture can reduce the production of pro-inflammatory cytokines [19]. Previous studies found that acupuncture treatment significantly decreased the serum levels of IL – 6 and TNF – α in patients with major depressive disorder. Acupuncture may suppress the activation of the nuclear factor – kappa B (NF-κB) signaling pathway in the brain of rats with depression – like behavior. The NF-κB pathway is a key regulator of inflammatory responses, and its inhibition by acupuncture may lead to a reduction in the production of inflammatory cytokines, thus exerting an anti-depressive effect [20]. We hypothesize that JTN treatment may modulate aberrant gene expression and DNA methylation patterns in the peripheral blood leukocytes of individuals with depression, potentially resulting in improved depressive symptoms. Therefore, the present study aims to (1) assess the efficacy and safety of JTN treatment for mild-to-moderate depression and (2) investigate the potential mechanisms underlying JTN's therapeutic effects in treating depression.

## Methods

### Trial design

This study is designed as a prospective, single-center, single-blind, randomized, controlled trial to evaluate the efficacy and safety of acupuncture (Jin's Three-Needle) for Mild-to-Moderate Depression. This study protocol conforms to the Consolidated Standards of Reporting Trials (CONSORT 2025) guidelines [21], the Standard Protocol Items: Recommendations for Interventional Trials (SPIRIT2025)(See S1 in the Supplementary file in S1 File) [22], and the Standards for Reporting Interventions in Controlled Trials of Acupuncture (STRICTA) (See S2 in the Supplementary file in S1 File) [23]. The flowchart of the trial design is depicted in Fig 1.

The trial will be conducted in accordance with the Declaration of Helsinki and has been approved by the Ethics Committee of Guangzhou Huangpu District Hospital of Traditional Chinese Medicine (TCMHHP2024120401) and registered with the International Traditional Medicine Clinical Trial Registry (ITMCTR2024000872). This trial is currently recruiting patients, which started on 01 January 2025 and is anticipated to end on 12 December 2026.

### Setting

The trial will be conducted in the outpatient department of Guangzhou Huangpu District Hospital of Traditional Chinese Medicine

### Participants

Participants will be recruited through advertisements at Guangzhou Huangpu District Hospital of Traditional Chinese Medicine. Only those participants who meet all inclusion criteria will be recruited for the study after they sign the informed consent form.

### Inclusion criteria

Participants who meet the following criteria will be included.

① Patients who fit the diagnostic criteria for depression in the Diagnostic and Statistical Manual of Mental Disorders(DSM-5);

② Patients with a HAMD-17score between 8 and 25.

③ Patients between the ages of 18 and 65 years;

④ Patients are voluntary and able to sign informed consent.

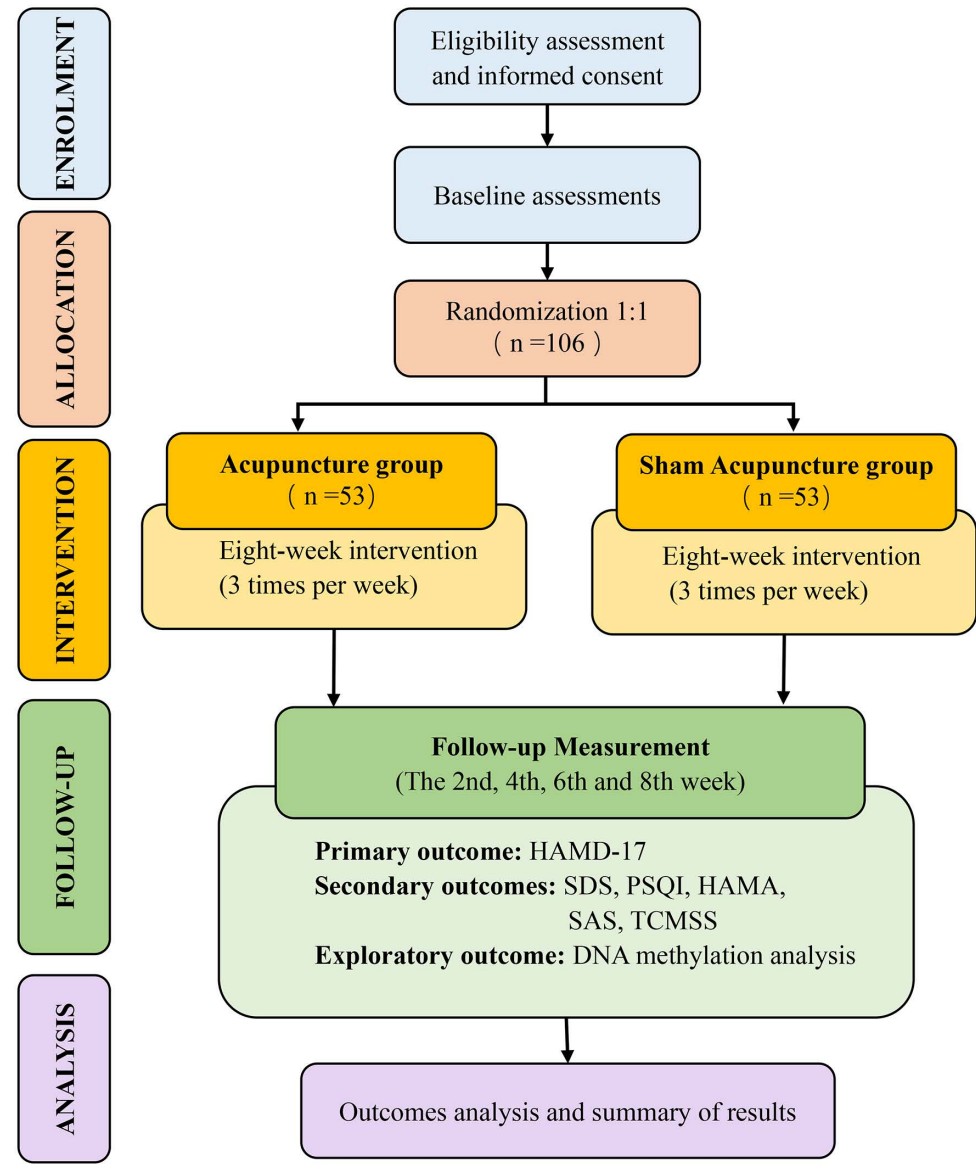

**Fig 1. Flowchart of trial procedures.** Notes: HAMD-17: The 17-item Hamilton Depression Rating Scale; SDS: Self-rating depression scale; PSQI, Pittsburgh sleep quality index; TCMSS: The traditional Chinese medicine syndrome score scale; HAMA: Hamilton Anxiety Scale; SAS: Self-Rating Anxiety Scale.

## Exclusion criteria

① Patients having suicidal tendencies (HAMD third score greater than 2 points), bipolar depression, and treatment-resistant depressions;

② Patients having immediate family or personal history of mental illness;

③ Patients who have serious physical health (as assessed by a doctor);

④ Patients who are pregnant or planning to become pregnant;

⑤ Patients who have taken antidepressants or receive acupuncture treatment within three months.

⑥ Patients with contraindications to acupuncture, including: a. Coagulation disorders (e.g., hemophilia) or current use of anticoagulant medication (e.g., warfarin); b. Severe thrombocytopenia; c. Local skin infections, ulcers, or scars at proposed needling sites; d. Presence of an implanted electronic medical device (e.g., pacemaker, spinal cord stimulator); e. Known severe cardiovascular conditions (e.g., uncontrolled arrhythmia).

## Randomization and allocation concealment

Researchers who are unaware of the trial will use SAS statistical software (SAS Inc., NCSU, version: 9.4, USA) to generate random numbers. The randomization sequence will be created using permuted blocks with a 1:1 allocation ratio (block size = 4) through a computer-generated process. The block size shall not be revealed until the end of the trial. Participants will be informed that they have an equal chance to be assigned to the acupuncture (Jin's Three-Needle) group or sham acupuncture group.

All the randomization number will be placed in opaque, sealed envelopes. After eligible participants sign the informed consent form and complete baseline assessments, they will receive a sealed envelope with the assigned number. Allocation concealment will be strictly implemented through an independent procedure conducted by an independent researcher at the clinical site, maintaining strict separation between randomization and outcome evaluation processes.

## Blinding

Acupuncturists will not be blinded to group allocation. However, the two groups of participants will avoid getting communicating with each other after randomization, and will receive treatment in separate treatment rooms to minimize subjective effects. The outcome assessors, data collectors, and statisticians will be blinded to group allocations. An independent researcher will supervise the conversation between participants and acupuncturists to avoid them communicating issues related to the allocation. At mid-study and end-study, participants will be asked via anonymous questionnaires to guess their assigned group and provide reasons. Statistical analysis of the results will be used to check whether participants remained blinded. If a significant number guess correctly, potential causes will be analyzed and measures will be adjusted. After each assessment, assessors will reassess the group allocations of randomly selected cases without access to their prior assessment results, while their decision-making processes will be recorded. The comparison between initial and repeated judgments will be used to assess whether assessors will be influenced.

## Intervention

The participants will be randomly assigned either to the acupuncture (Jin's Three-Needle) group or the sham acupuncture group. All acupuncture sessions will be performed by acupuncturists from China who are licensed as acupuncturists with more than 3 years of experience in clinics.

Before initiation of the study, the acupuncturists will participate in training sessions to ensure that the skills of the acupuncturists are consistent and receive a brochure that shows the acupuncture manipulation with detailed information. Patients in both groups will undergo 24 acupuncture sessions over the course of 8 weeks (three treatment per week, 30 min per session). Our study will utilize a disposable sterile needle (Hwato, Suzhou, China; lengths and diameters 0.3 mm × 25 mm or 0.3 mm × 40 mm). For the location of acupuncture points, we will refer to the WHO Standard Acupuncture Point Locations in the Western Pacific Region [24]. The schedule of the trial is described in detail in Fig 2.

**Acupuncture (Jin's Three-Needle) group.** According to the traditional theory of Jin's Three-Needle and our previous studies [12,25], acupuncture interventions were developed by a consensus of acupuncture experts. The

| | TRIAL PERIOD | | | | | | | | | | |
|---|---|---|---|---|---|---|---|---|---|---|---|
| | Enrolment | Post-Randomizasion | | | | | | | | | Close out |
| | | Intervention | | | | | | | | | |
| **Timepoint (week)** | −1 | 0 | 1 | 2 | 3 | 4 | 5 | 6 | 7 | 8 | >8 |
| **ENROLLMENT:** | × | | | | | | | | | | |
| Eligibility screen | × | | | | | | | | | | |
| Informed consent | × | | | | | | | | | | |
| Randomizasion | | × | | | | | | | | | |
| **INTERVENTIONS:** | | | | | | | | | | | |
| Acupuncture group | | | ×◆—————————————————▶× | | | | | | | | |
| Sham Acupuncture group | | | ×◆—————————————————▶× | | | | | | | | |
| **ASSESSMENTS:** | | | | | | | | | | | |
| HAMD-17 | | × | | × | | × | | × | | × | |
| SDS | | × | | × | | × | | × | | × | |
| PSQI | | × | | × | | × | | × | | × | |
| TCMSS | | × | | × | | × | | × | | × | |
| HAMA | | × | | × | | × | | × | | × | |
| SAS | | × | | × | | × | | × | | × | |
| Blood sample | | × | | | | | | | | × | |
| **Adverse event** | | | × | × | × | × | × | × | × | × | |
| **Analysis of outcomes** | | | | | | | | | | | × |

**Fig 2. Schedule of enrolment, interventions, and assessments.** Notes: HAMD-17: The 17-item Hamilton Depression Rating Scale; SDS: Self-rating depression scale; PSQI, Pittsburgh sleep quality index; TCMSS: The traditional Chinese medicine syndrome score scale; HAMA: Hamilton Anxiety Scale; SAS: Self-Rating Anxiety Scale.

acupoints selected were "Yusan-zhen, "Dingshen-zhen", and "Zusan-zhen", which are commonly used to treat psychiatric disorders. Baihui (GV20), Sishen-I(GV21), Sishen-II(GV19), Sishen-III, Sishen-IV, Dingshen-I, Dingshen-II, Dingshen-III, Neiguan(PC6), Zusanli(ST36), Sanyinjiao(SP6), and Taichong(LR3) acupoints will be used in the acupuncture treatment.

PC6, ST36, SP6 and LR3 will be needled at an angle of 45–90° to the participant's skin, while Dingshen-zhen and Sishen-zhen will be needled at an angle of 15–30° to the skin. The needles will be inserted at a depth of 5–30 mm and the needles will be stimulated by manual rotation to achieve the typical acupuncture sensation of de qi, which is characterized as soreness, numbness and heaviness. Needles will be retained for 30 min and manually stimulated every 10 minutes. The location of acupuncture points is described in Table 1 and Fig 3.

**Sham acupuncture group.** The sham acupuncture consisted of sham acupoint, shallow needle insertion using thin and short needles at non acupuncture points [26]. Sham acupoints are systematically located near to the verum acupuncture sites. We identified non – meridian regions that are far from the verum acupoints and have no meridian associations as the new sham acupoint locations. (See S3 in the Supplementary file in S1 File). The needle insertion will be gentle, superficial and oblique <5mm, with 30-minute retention time. No needle manipulation will be performed throughout the intervention session. After the trial, all participants in sham acupuncture group will receive vouchers for 24 true acupuncture bonus sessions as compensation.

## Outcomes

**Primary outcome.** The primary outcome is the change in the 17-item Hamilton Depression Rating Scale (HAMD-17) score from baseline to the end of the 8-week treatment period (Week 8). The HAMD-17 is the most employed

**Table 1. Framework of the acupuncture point prescription.**

| Acupuncture points | Description |
|---|---|
| *Sishen-I(GV21)* | On the head, 3.5 B-cun superior to the anterior hairline, on the anterior median line. |
| *Sishen-II* (GV19) | On the head, 5.5 B-cun superior to the posterior hairline, on the posterior median line. |
| GV20: Baihui | On the head, 5B-cun superior to the anterior hairline, on the anterior median line. |
| *Sishen-III* | On the head, 1.5*cun* left lateral to the anterior median line and at the same level as GV20. |
| *Sishen-IV* | On the head, 1.5*cun* right lateral to the anterior median line and at the same level as GV20 |
| EX-HN3: Yintang | On the head, between the right medial end of the eyebrow and the left one. |
| *Dingshen-I* | On the head, directly 0.5*cun* superior to EX-HN3 |
| GB14: Yangbai | On the head, 1B-cun superior to the eyebrow, directly superior to the centre of the pupil. |
| *Dingshen-II* | On the head, directly 0.5*cun* superior to left GB14 |
| *Dingshen-III* | On the head, directly 0.5*cun* superior to right GB14 |
| PC6: Neiguan | On the anterior aspect of the forearm, between the tendons of the palmaris longus and the flexor carpi radialis, 2B-cun proximal to the palmar wrist crease. |
| ST36: Zusanli | On the anterior aspect of the leg, on the line connecting ST35 with ST41, 3 B-cun inferior to ST35. |
| SP6: Sanyinjiao | On the tibial aspect of the leg, posterior to the medial border of the tibia, 3 B-cun superior to the prominence of the medial malleolus. |
| LR3: Taichong | On the dorsum of the foot, between the first and second metatarsal bones, in the depression distal to the junction of the bases of the two bones, over the dorsalis pedis artery. |

The prescribed of acupoints are come from the WHO Standard Acupuncture Point Locations in the Western Pacific Region

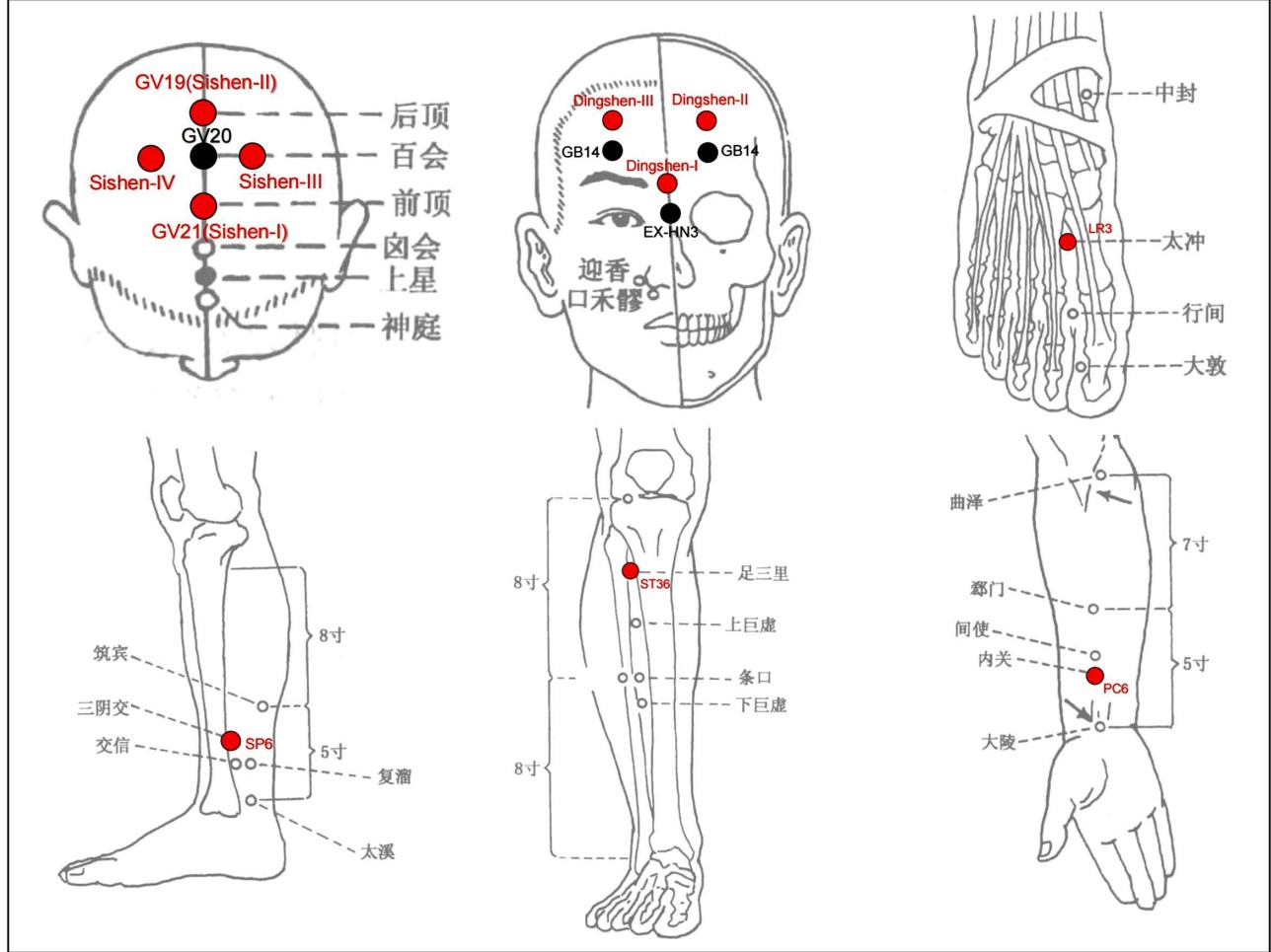

**Fig 3. Acupuncture points used in the study.**

classical clinical depression symptom rating scale, which measures five domains related to depression, including anxiety/somatization, weight, cognitive disturbance, psychomotor retardation and sleep disturbance [27]. HAMD-17 consists of 17 items that include 3 (scored from 0 to 2) and 5 (scored from 0 to 4) response variants, which reflect ascending symptom severity levels. Total scores are categorized as follows: 0–7 (normal), 8–24 (mild to moderate depressive disorder), and ≥25 (major depressive disorder).

**Secondary outcomes.** Secondary outcomes included scores on the Traditional Chinese Medicine Syndrome Score Scale (TCMSS), Pittsburgh Sleep Quality Index (PSQI), Self-rating depression scale (SDS); Hamilton Anxiety Scale (HAMA); Self-Rating Anxiety Scale (SAS).

(1) TCMSS: It consists of 23 items with 3 (scored 0–2), 4 (scored 0–3) and 4(scored 0, 2, 4 and 6) response variants that are scored according to increasing symptom intensity [28]. Higher scores are indicative of "Liver Qi Stagnation" and "Spleen Deficiency" patterns in traditional Chinese medicine, which reflect symptoms of depression. The TCMSS is interpreted as follows: < 40(mild/moderate depressive disorder) and ≥ 40(major depressive disorder).

(2) PSQI: It assesses sleep disturbances through seven domains: subjective sleep quality, sleep duration, sleep latency, habitual sleep efficiency, sleep disturbances, daytime dysfunction, and use of sleep medications [29]. Each component is rated on a 4-point Likert-type scale (0–3), with total scores ranging from 0 to 21. Higher scores reflect greater severity of sleep impairment, with a threshold score >5 indicating clinically significant sleep dysfunction.

(3) SDS: It developed by Dr. William W.K. Zung in 1965, is a psychometric instrument comprising 20 items designed to assess subjective depressive experiences [30]. It employs a 4-point Likert-type scoring system, wherein participants rate item applicability based on symptom occurrence frequency: 1 = Never or rarely; 2 = Occasionally; 3 = Frequently; 4 = Consistently. Higher total scores indicate greater symptom severity, with clinical stratification as follows: 53–62: Mild depression (clinically significant); 63–72: Moderate depression; ≥ 73: Severe depression.

(4) HAMA: It developed by Max Hamilton in 1959, is a clinician-administered psychometric instrument originally designed for psychiatric practice [31]. HAMA consists of 14 items, with all items scored on a 0–4 scale using a 5-point rating system. The criteria for each level are as follows: 0 points indicate no symptoms; 1 point indicates mild symptoms; 2 points indicate moderate symptoms; 3 points indicate severe symptoms; and 4 points indicate extremely severe symptoms. The higher the total score, the more severe the anxiety symptoms. Specifically, a total score of ≥29 points may indicate severe anxiety; ≥ 21 points indicate definite significant anxiety; ≥ 14 points indicate definite anxiety; a score above 7 may suggest anxiety; and a score below 7 indicates no anxiety symptoms.

(5) SAS: It developed by Dr. William W.K. Zung in 1971, is a psychometric instrument comprising 20 items designed to assess subjective anxiety [32]. It employs a 4-point Likert-type scoring system, wherein participants rate item applicability based on symptom occurrence frequency: 1 = Never or rarely; 2 = Occasionally; 3 = Frequently; 4 = Consistently. Higher SAS scores generally reflected more serious levels of anxiety concerns. The cutoff point of SAS score 50 considered as the presence of clinical anxiety concern in Chinese population, with severity stratification as follows: 50–59: Mild anxiety; 60–69: Moderate anxiety; ≥ 70: Severe anxiety.

**Safety.** In the event of adverse events (AEs), corrective measures and prognostic interventions will be implemented commensurate with the severity of the event. All AEs must be documented, addressed within 24 hours, and subjected to objective, evidence-based evaluation utilizing adverse reaction data. Concurrently, a structured investigation will be conducted to determine whether the AEs are treatment-related, evaluate the therapeutic interventions associated with the events, and assess the potential benefits of modifying the treatment protocol within the trial framework. For minor acupuncture-related AEs (e.g., dizziness, needle stagnation, nausea, hematoma formation, localized infection), participants may continue participation in the early-phase intervention trial only following reconfirmation of informed consent and rigorous exclusion of confounding factors. Participants experiencing severe AEs deemed causally linked to the trial intervention will be immediately withdrawn from the study and provided with appropriate medical treatment in accordance with ethical and clinical guidelines. All acupuncturists and research assessors will be trained in recognizing signs of clinical deterioration and are aware of the emergency referral procedures. Participants who experience significant worsening of their depression will be offered, in consultation with their treating physician, the option to initiate standard antidepressant medication. This would be considered a major protocol deviation and lead to study withdrawal, but their data would continue to be followed for safety analyses.

## Quality control, data management and monitoring

Prior to participant recruitment, all team members will be required to complete a standardized training workshop to ensure strict adherence to the study protocol and comprehensive understanding of trial administration procedures. The interventions will be performed by licensed acupuncturists with more than 3 years of experience in hospital. The acupuncturists will also receive a brochure about the protocol and standard operating procedures of acupuncture (Jin's Three-Needle).

All data will be systematically collected and recorded in case report forms (CRFs) with chain-of-custody documentation. All data will be entered into a password-protected computer by research personnel unaware of the group assignments. The data will then be double-checked by investigators following data entry. Data quality will be regularly monitored by research assistants and overseen by monitors. The original CRFs and all other forms will be archived securely at the Clinical Research Center of Guangzhou Huangpu District Hospital of Traditional Chinese Medicine.

The Clinical Research Center of Guangzhou Huangpu District Hospital of Traditional Chinese Medicine will be responsible for several key aspects of this trial: monitoring the data, managing regulatory compliance, conducting independent audits every three months without influence from investigators or the sponsor, and determining if any premature closure of the study is necessary.

### Sample size calculation

The sample size was determined by utilizing the HAMD-17 scale score as the evaluative index. Based on previous study [33], after 8 weeks of treatment, the HAMD-17 score difference between the baseline and the endpoint in the acupuncture group should be approximately $10.79 \pm 1.40$, while the sham acupuncture group should be approximately $9.82 \pm 1.80$. Type 1 error is assumed at 0.05, and type 2 error is assumed at 0.2. PASS11.0 software (NCSS Statistical Software, Kaysville, UT, USA) has been used to determine the sample size, and the minimum sample size is 45 subjects for each group. Considering a dropout rate of 15%, a total of 106 subjects will be required with 1:1 allocation to each group (53 participants per group) for this study.

### Statistical analysis

Intention-to-treat (ITT) will be used for the complete follow-up analysis. All randomized participants who received at least one treatment session and for whom at least one post-baseline assessment will be included for the full analysis in the full analytic dataset (FAS). Missing data will be replaced according to the principle of multiple imputations. Participants demonstrating strict protocol adherence and completing the full treatment course will be included in the Per Protocol Set (PPS) for efficacy analysis. All randomized participants who received at least one treatment will be included in the Safety Analysis Set (SS).

Data management will be performed using Epidata software (v3.1, Odense, Denmark), with subsequent analyses conducted in IBM SPSS Statistics 25.0 (IBM Corp, Version 25.0, New York, USA). Baseline characteristics will be analyzed using FAS, while efficacy outcomes will be evaluated through both FAS and PPS. Safety assessments for acupuncture (Jin's three-needle) intervention will be utilized SS. Continuous data will be statistically described using mean ± standard deviation and qualitative data will be provided as percentages or proportions, along with confidence intervals of 95%. Outcomes with multiple time points will be analyzed using repeated-measures analysis of variance (ANOVA), with treatment group and follow-up time as fixed effects and individual as a random effect, to compare the effect sizes at each time point. We plan to use the Shapiro – Wilk test to assess the normality of the residuals in the repeated measures ANOVA model. If the normality assumption is violated, we will consider appropriate non – parametric alternatives, such as the generalized estimating equations (GEE) approach. For between-group comparisons, normally distributed data with homogeneous variances will be analyzed using the two independent samples t-test. Non-normally distributed continuous variables or those with heterogeneous variances will be analyzed using the Kruskal-Wallis H test. All Statistical tests will be based on two-sided tests, with a significance level of $\alpha = 0.05$ for statistically significant results. The Bonferroni test will be used for post hoc pairwise comparisons. We plan to conduct a sensitivity analysis to assess the impact of varying the assumed value of the Minimal Clinically Important Difference (MCID) (within a reasonable range based on existing literature and expert opinion) on the study findings and conclusions.

We will investigate whether acupuncture (Jin's Three-Needle) regulates aberrant gene expression and DNA methylation patterns in the peripheral blood leukocytes of patients with mild-to-moderate depression. Details on molecular analysis are provided in S4 of the Supplementary File in S1 File.

The final statistical analyses will be conducted by an independent data analyst who is blinded to the treatment group allocation.

## Discussion

JTN treatment for patients with mild-to-moderate depression has not been thoroughly investigated. This randomized controlled trial aims to (1) evaluate the efficacy and safety of JTN treatment for mild-to-moderate depression and (2) examine the underlying mechanisms of its therapeutic effects in treating depression.

Individuals with mild-to-moderate depression typically exhibit fewer and less severe depressive symptoms. Often, there are no observable signs of low mood or psychomotor retardation. However, mild-to-moderate depression can easily progress to major depressive disorder. Major depression significantly impacts patients' physical health, social functioning, and daily activities, representing a condition with a high disease burden [34]. Therefore, it is crucial for clinicians and researchers to develop effective interventions to manage mild-to-moderate depression and prevent its progression to more severe forms.

Acupuncture is widely applied in clinical practices across China. The efficacy of acupuncture is primarily determined by the careful selection and strategic combination of acupuncture points and standardized procedures. In our study, the JTN treatment was thoroughly developed based on TCM theory and expert acupuncturists' clinical experience, which has been practiced for decades. A key characteristic of JTN lies in its unique approach to acupoint selection. The JTN treatment protocol incorporates multiple acupoint groups, with each group comprising three to four specialized acupoints carefully selected according to local points, influential points, or empirical points. JTN demonstrates therapeutic efficacy in treating mental disorders and assisting patients in modulating their mental states. In China, acupuncture methods are widely utilized by practitioners to manage depression. Numerous clinical trials have shown that it might effectively alleviate patients' symptoms, highlighting its therapeutic validity and clinical relevance [35]. Several studies have demonstrated that acupuncture may not only serve as a safe adjunctive treatment combined with antidepressants but also exhibit greater efficacy in alleviating depressive symptoms [36–38]. Unlike conventional acupuncture, JTN integrates local, influential, and empirical points to achieve layered therapeutic effects on somatic, emotional, and energetic levels. For instance, the combination of HT7 (Shenmen), ST36 (Zusanli), and GB20 (Fengchi) in JTN simultaneously targets serotonin regulation, immune modulation, and vagal nerve activation [39]. Lin, W., et al. suggest JTN may offer superior symptom relief and lower tolerance risks compared to standard acupuncture [40], positioning it as a promising intervention for preventing depression progression. As evidenced by previous studies, JTN has been widely employed to alleviate symptoms of depression and anxiety, as well as enhance sleep quality [41]. Although growing evidence indicated that acupuncture was a very promising treatment for depression, there was still insufficient evidence [42] to determine the efficacy of acupuncture due to lacking well-designed studies. Therefore, we will rigorously adhere to the SPIRIT statement and STRICTA recommendations, standardizing our study protocol to elucidate the underlying mechanisms of acupuncture (Jin's Three-Needle) in treating mild-to-moderate depression.

Our study design incorporates safeguards to minimize potential biases. However, several limitations merit consideration. First, the single-center approach lacks regional diversity. Second, due to the lack of a planned post-intervention follow-up period, the long-term outcomes and durability of treatment effects could not be assessed. Third, depression diagnosis relies on subjective assessments without validated biomarkers, potentially introducing selection and measurement biases. Fourth, practitioner blinding was impossible given the intervention's technical requirements, creating a risk of performance bias. We are currently working to standardize this trial's procedure.

In conclusion, the results of this trial will provide evidence regarding the effectiveness and safety of JTN treatment for patients with mild-to-moderate depression and offer high-quality, clinically relevant evidence to inform the design of future larger-scale randomized controlled trials.

## Supporting information

**S1 File.**

(PDF)

## Author contributions

**Conceptualization:** Hao Wen, Chong Zhang.

**Data curation:** Jiafeng Wang, Kaiying Zhang.

**Formal analysis:** Xingyu Xiang.

**Investigation:** Jiafeng Wang, Na Zhang, Yiqing Shan, Xiaolan Yang, Pengcheng Zhang, Xingyu Xiang.

**Project administration:** Na Zhang, Yiqing Shan.

**Writing – original draft:** Hao Wen.

**Writing – review & editing:** Baoguo Zhang, Chong Zhang.

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
