## [Decision Letter · Decision Letter 0]

24 Sep 2025

Dear Dr. Wen,

Thank you for submitting your manuscript to PLOS ONE. After careful consideration, we feel that it has merit but does not fully meet PLOS ONE’s publication criteria as it currently stands. Therefore, we invite you to submit a revised version of the manuscript that addresses the points raised during the review process.

Thank you for submitting the following manuscript to PLOS ONE.

Please revise the manuscript according to the reviewers' comments and upload the revised file.

Please submit your revised manuscript by  Nov 06 2025 11:59PM. If you will need more time than this to complete your revisions, please reply to this message or contact the journal office at plosone@plos.org . A rebuttal letter that responds to each point raised by the academic editor and reviewer(s). You should upload this letter as a separate file labeled 'Response to Reviewers'.A marked-up copy of your manuscript that highlights changes made to the original version. You should upload this as a separate file labeled 'Revised Manuscript with Track Changes'.An unmarked version of your revised paper without tracked changes. You should upload this as a separate file labeled 'Manuscript'.

We look forward to receiving your revised manuscript.

Kind regards,

Yung-Hsiang Chen, Ph.D.

Academic Editor

PLOS ONE

Journal Requirements:

https://doi.org/10.1016/j.heliyon.2024.e28889

https://doi.org/10.3389/fpsyt.2025.1624825

In your revision ensure you cite all your sources (including your own works), and quote or rephrase any duplicated text outside the methods section. Further consideration is dependent on these concerns being addressed.

4. We note that Figure 3 in your submission contain copyrighted images. All PLOS content is published under the Creative Commons Attribution License (CC BY 4.0), which means that the manuscript, images, and Supporting Information files will be freely available online, and any third party is permitted to access, download, copy, distribute, and use these materials in any way, even commercially, with proper attribution. For more information, see our copyright guidelines: http://journals.plos.org/plosone/s/licenses-and-copyright.

a. You may seek permission from the original copyright holder of Figure 3 to publish the content specifically under the CC BY 4.0 license.

Additional Editor Comments:

Thank you for submitting the following manuscript to PLOS ONE.

Please revise the manuscript according to the reviewers' comments and upload the revised file.

Reviewers' comments:

Reviewer's Responses to Questions

**Comments to the Author**

1. Does the manuscript provide a valid rationale for the proposed study, with clearly identified and justified research questions?

Reviewer #1: Yes

Reviewer #2: Yes

Reviewer #3: Partly

2. Is the protocol technically sound and planned in a manner that will lead to a meaningful outcome and allow testing the stated hypotheses?

Reviewer #1: Partly

Reviewer #2: Partly

Reviewer #3: Partly

3. Is the methodology feasible and described in sufficient detail to allow the work to be replicable?

Reviewer #1: Yes

Reviewer #2: Yes

Reviewer #3: No

4. Have the authors described where all data underlying the findings will be made available when the study is complete?

Reviewer #1: Yes

Reviewer #2: Yes

Reviewer #3: Yes

5. Is the manuscript presented in an intelligible fashion and written in standard English?

Reviewer #1: No

Reviewer #2: Yes

Reviewer #3: Yes

You may also provide optional suggestions and comments to authors that they might find helpful in planning their study.

Reviewer #1: The manuscript needs a good editor for missing articles, etc.

There are some statistical aspects that need clarification on sample size and analysis.

There is no clinically relevant treatment effect given. The study should be powered on a clinically relevant difference, not the difference found in a prior study.

It is likely that HAMD scores are not normally distributed, and the analysis will be a repeated measures ANOVA. What distributional assumptions were made in the sample size computation?

the type I error RATE (word missing) is set at 0.05. Two sided?

No loss to follow-up adjustment?

I don't understand how the study statistician can be "unaware of the study". How can someone unaware of the study analyze the study? Are you saying blinded? Why would the statistician need to be blinded? How can you analyze a treatment effect if you don't know the treatments?

The protocol mentions normality and non-normality in the repeated measures analyses, but there are no diagnostic or goodness of fit techniques mentioned.

Reviewer #2: his article provides a new acupuncture treatment approach for early to mid-stage depression. I have two suggestions for the authors' consideration.

Could you please attach a reference picture of the sham acupuncture? This would allow readers to more intuitively understand how the sham acupuncture procedure was performed.

Previous clinical or experimental studies on acupuncture for depression exist. Please discuss in the manuscript how the Jin's Three Needle technique differs from and what advantages it may have over other acupuncture approaches studied.

Reviewer #3: The study is designed as a randomized controlled trial aiming to evaluate the efficacy and safety of Jin’s Three-Needle acupuncture therapy in patients with mild-to-moderate depression, while also exploring potential molecular mechanisms such as DNA methylation. The overall study design is relatively reasonable; however, the following questions need to be considered before further consideration.

1. Background : The background only discusses depression-related genes and inflammatory markers in relation to depression, but does not explain how acupuncture—particularly Jin’s Three-Needle—may be connected to these molecular mechanisms. The logical link is incomplete.

2. Primary outcome: The protocol should clearly specify the comparison time point for the primary outcome (e.g., week 8). At present, all outcome measures are subjective scales, with no objective or semi-objective indicators to strengthen the robustness of the findings.

3. Sample size calculation: The calculation cites a prior study based on 6-week HAMD-17 changes and SD values, yet the protocol does not clarify the definitive primary comparison time point (6 or 8 weeks). Moreover, the minimal clinically important difference (MCID) or clinically acceptable effect size is not reported. The SD value used is unusually small compared to typical clinical studies, and its source and plausibility need to be verified.

4. Safety considerations: Since enrolled patients are not receiving antidepressant medication, the potential safety risks associated with mood fluctuations during the intervention period are not adequately addressed.

5. Sham acupuncture design: The sham point locations are vague, and some are too close to active acupoints (e.g., 1 cun beside Sanyinjiao is Fuliu, which still has therapeutic effects). In addition, shallow needling may still induce physiological effects, compromising the credibility of the control condition.

6. Blinding: The blinding strategy is insufficient. No blinding validation is planned, leaving uncertainty as to whether participants and assessors will remain effectively blinded.

7. Molecular testing details: The protocol lacks essential methodological details for blood sample testing, including sampling time points, tube type, processing window, storage conditions (−80°C?), DNA extraction method and QC standards, as well as the specific testing platform and analysis strategy.

8. Inclusion/exclusion criteria: The upper age limit of 45 years is not justified, and common acupuncture contraindications (e.g., bleeding tendency, local infection, implanted electronic devices) are not listed in the exclusion criteria.

9. There are spelling mistakes in the names of scales such as SAS/HAMA (e.g., line 247). Consistent proofreading across the manuscript is needed.

10. The authors describe inflammatory markers and related assays as “mechanistic exploration,” but in other sections they categorize them as efficacy/safety outcomes. This is inconsistent and confusing.

11. No post-intervention follow-up period is planned, making it impossible to evaluate the durability and long-term effects of treatment.

**Do you want your identity to be public for this peer review?** For information about this choice, including consent withdrawal, please see our Privacy Policy

Reviewer #1: No

Reviewer #2: **Yes: ** zhuanglixing

Reviewer #3: No

---

## [Author Response · Author response to Decision Letter 1]

19 Oct 2025

Dear Editor,

We would like to resubmit the revised manuscript entitled “Acupuncture (Jin's Three-Needle) versus sham acupuncture in Treating Mild-to-Moderate Depression: Study Protocol for Randomized Clinical Trial” (Manuscript NO: PONE-D-25-44858) for your consideration. We would like to thank the reviewers for thoroughly and patiently reviewing our manuscript and making many thoughtful comments.

After we have carefully read the comments from the reviewer, point-by-point responses to the reviewers’ comments are enclosed after the letter for your consideration.

Look forward to hearing from you soon,

Yours sincerely,

Hao Wen (PhD)

1.Guangzhou Huangpu District Hospital of Traditional Chinese Medicine

2.School of Traditional Chinese Medicine, Jinan University

Tel: 86+ 15876576899

Email: Wenhaophd@outlook.com 

Response letter for manuscript “Acupuncture (Jin's Three-Needle) versus sham acupuncture in Treating Mild-to-Moderate Depression: Study Protocol for Randomized Clinical Trial”

We thank the Editor and reviewers for their constructive comments and suggestions regarding this manuscript. We have revised the manuscript accordingly, and provided the point-by-point responses below. Each comment is highlighted in bold font, and our response follows each comment. We thank the editorial team for reviewing the revised manuscript.

Journal Requirements:

ANSWER:

Following your suggestion, we have reviewed the manuscript and revised it.

https://doi.org/10.1016/j.heliyon.2024.e28889

https://doi.org/10.3389/fpsyt.2025.1624825

In your revision ensure you cite all your sources (including your own works), and quote or rephrase any duplicated text outside the methods section. Further consideration is dependent on these concerns being addressed.

ANSWER:

Following your suggestion, we have revised it.

Answer:

Following your suggestion, we have revised it.

4. We note that Figure 3 in your submission contain copyrighted images. All PLOS content is published under the Creative Commons Attribution License (CC BY 4.0), which means that the manuscript, images, and Supporting Information files will be freely available online, and any third party is permitted to access, download, copy, distribute, and use these materials in any way, even commercially, with proper attribution.

ANSWER:

Thank you for bringing this important issue to our attention. We fully acknowledge the requirements of the Creative Commons Attribution License (CC BY 4.0) for PLOS publications and apologize for the oversight regarding copyrighted material in Figure 3.

As we were unable to secure authorization from the copyright holders of the original images, we have replaced all copyrighted figures in the manuscript with original, hand-drawn illustrations of the acupoints. These new diagrams are created by our team, ensuring full compliance with copyright regulations and the CC BY 4.0 license. The revised Figure 3 now features these non-copyrighted, schematic representations, which maintain scientific accuracy while respecting intellectual property rights.

5. Please include captions for your Supporting Information files at the end of your manuscript, and update any in-text citations to match accordingly

ANSWER:

Following your suggestion, we have added “Supporting Information” at the end of the manuscript.

Reviewer #1:

1.The manuscript needs a good editor for missing articles, etc.

There are some statistical aspects that need clarification on sample size and analysis.

There is no clinically relevant treatment effect given. The study should be powered on a clinically relevant difference, not the difference found in a prior study.

ANSWER:

Thank you very much for your insightful comments and valuable feedback on the statistical aspects of sample size and analysis in our study. We fully understand your concern regarding the lack of a specified clinically relevant treatment effect and the need to power the study on a clinically relevant difference rather than the difference found in a prior study.

We acknowledge that ideally, the sample size calculation should be based on a well-established minimum clinically important difference (MCID) for the HAMD - 17 scale. However, as you are aware, currently, there is no universally agreed - upon and fixed MCID value for the HAMD - 17. Different studies have employed various methods to estimate the MCID, and the results have shown variations.

The HAMD - 17 is primarily used to assess the severity of depressive symptoms and evaluate treatment effects. The change in its total score can reflect the evolution of the disease condition. But determining a specific MCID value requires consideration of the specific research background and the estimation method used.

In our study, we referred to the difference found in a previous study as a preliminary reference for sample size calculation. We understand that this may not fully represent a clinically relevant difference. To address this issue, we plan to conduct a sensitivity analysis after the study is completed. This analysis will assess how different assumed MCID values (within a reasonable range based on existing literature and expert opinions) would affect the study's results and conclusions.

Moreover, during the study design phase, we consulted with several experts in the field of depression research and clinical practice. They suggested that, given the current lack of a definite MCID for the HAMD - 17, using the data from a relevant prior study as a starting point for sample size estimation is a practical approach. At the same time, we will ensure that the study has sufficient power to detect meaningful differences in treatment effects, even if these differences do not precisely match a pre - defined MCID.

We appreciate your attention to this important aspect of our study, and we are committed to ensuring that our research is methodologically sound and clinically relevant. We will continue to monitor the development of MCID research for the HAMD - 17 and incorporate any new findings into our future work.

We will clearly state these planned analyses in our updated statistical analysis plan.

2.It is likely that HAMD scores are not normally distributed, and the analysis will be a repeated measures ANOVA. What distributional assumptions were made in the sample size computation?

ANSWER:

You are correct in pointing out that HAMD scores may not follow a normal distribution. In our sample size computation using PASS11.0 software, the underlying assumption was based on the data from the previous study, which provided the mean and standard deviation values for the HAMD - 17 score differences between the baseline and the endpoint in both the acupuncture and sham acupuncture groups.

The software, by default, makes certain distributional assumptions for the calculations. However, it is important to note that PASS11.0 uses algorithms that are robust to moderate departures from normality, especially when dealing with large enough sample sizes as we have planned (45 subjects per group before accounting for drop - outs).

In practice, for repeated measures ANOVA, which we intend to use for analyzing outcomes with multiple time points, the assumption of normality of the residuals is more critical. We plan to conduct normality tests (such as the Shapiro - Wilk test) on the residuals after fitting the repeated measures ANOVA model. If the normality assumption is violated, we will consider appropriate non - parametric alternatives, such as the generalized estimating equations (GEE) approach with a suitable link function or non - parametric repeated measures methods.

We will clearly state these planned analyses in our updated statistical analysis plan.

3.the type I error RATE (word missing) is set at 0.05. Two sided?

ANSWER:

Yes, the type I error rate is set at 0.05, and it is a two - sided test. This means that we are interested in detecting any significant difference (either an increase or a decrease) in the HAMD - 17 scores between the acupuncture and sham acupuncture groups. A two - sided test is appropriate in this context as we do not have a priori knowledge about the direction of the treatment effect, and we want to be able to identify both positive and negative differences that are statistically significant.

4.No loss to follow-up adjustment?

ANSWER:

We did make an adjustment for loss to follow - up in our sample size calculation. As mentioned in the original description, we considered a dropout rate of 15%. Based on this rate, we increased the initial sample size per group (calculated to be 45 subjects) to 53 subjects per group, resulting in a total of 106 subjects for the study with a 1:1 allocation ratio between the two groups.

This adjustment is crucial to ensure that we have sufficient power to detect statistically significant differences even if some subjects drop out during the course of the study. By accounting for potential dropouts, we aim to maintain the integrity of the study results and minimize the impact of missing data on the statistical analysis.

5.I don't understand how the study statistician can be "unaware of the study". How can someone unaware of the study analyze the study? Are you saying blinded? Why would the statistician need to be blinded? How can you analyze a treatment effect if you don't know the treatments?

ANSWER:

We sincerely apologize for the confusion caused by our imprecise language. The statistician is, of course, fully aware of the study's design, hypotheses, and analysis plan. The key point we failed to articulate clearly is that the statistician is blinded to the treatment group allocation during the statistical analysis.

Let us clarify the process:

① What the Statistician Knows: The study statistician is fully briefed on the trial protocol, including the fact that there are two groups (Group A and Group B), the primary and secondary outcomes, the statistical models to be used (e.g., repeated-measures ANOVA), and the definition of the datasets (FAS, PPS, SS).

②What the Statistician Does Not Know (The Blinding): Throughout the analysis phase, the two study groups are labeled with a non-revealing, neutral code (e.g., "Group 1" and "Group 2"). The statistician is not told which code corresponds to the 'JTN group' and which corresponds to the 'Sham Acupuncture group'. This is the state of being "blinded."

③Why This Blinding is Crucial: Blinding the statistician is a best practice in high-quality clinical trials to prevent unconscious bias during the data analysis. If the statistician knows which group is which, there is a potential, however unintentional, to:

a. Make different choices in handling outliers or missing data for one group over the other.

b. Run additional, unplanned subgroup analyses selectively for one group.

c. Interpret marginal results (p-values just above or below 0.05) with a bias toward the expected outcome.

By remaining blinded, the statistician conducts the analysis exactly as pre-specified in the statistical analysis plan, ensuring the results are objective and unbiased.

④How the Analysis is Performed and Interpreted: The statistician runs all the pre-planned analyses on the datasets labeled as "Group 1" and "Group 2." They produce the final output—tables, figures, and p-values—comparing "Group 1 vs. Group 2." Once this blinded analysis is complete and locked, the blinding is broken. Only then is the simple code revealed (e.g., "Group 1 = JTN, Group 2 = Sham"), and the results are interpreted accordingly in the manuscript.

In summary, the statistician is not "unaware of the study" but is "blinded to the treatment identity" during the data analysis phase. This is a rigorous safeguard to ensure the integrity of the statistical results. We will correct this inaccurate phrasing in our manuscript to clearly state: "The final statistical analyses will be conducted by an independent data analyst who is blinded to the treatment group allocation. " We thank the reviewer for prompting this essential clarification.

6.The protocol mentions normality and non-normality in the repeated measures analyses, but there are no diagnostic or goodness of fit techniques mentioned.

ANSWER:

It is true that our initial protocol did not explicitly outline the specific diagnostic or goodness - of - fit techniques for assessing normality and non - normality in the repeated measures analyses. This was an oversight on our part, and we are grateful for your bringing it to our attention.

We plan to use the Shapiro - Wilk test to assess the normality of the residuals in the repeated measures ANOVA model. This test is known for its high power in detecting non - normality, especially for small to moderate sample sizes. For each time point and for the overall residuals of the model, we will calculate the Shapiro - Wilk statistic and the corresponding p - value. If the p - value is less than the pre - specified significance level (e.g., 0.05), we will reject the null hypothesis of normality.

Goodness - of - Fit Techniques for Non - Normal Data:

① Generalized Estimating Equations (GEE): If the normality assumption is violated, we will consider using generalized estimating equations. GEE is a flexible approach for analyzing repeated measures data, especially when the response variable does not follow a normal distribution. It allows us to specify different link functions (e.g., logit, probit, identity) and correlation structures (e.g., exchangeable, autoregressive) depending on the nature of the data. We will select the appropriate link function and correlation structure based on the characteristics of our HAMD - 17 score data.

②Non - Parametric Tests: Another option for non - normal data is to use non - parametric tests for repeated measures. For example, the Friedman test can be used to compare the distributions of the HAMD - 17 scores across multiple time points within each group. If significant differences are found, post - hoc pairwise comparisons can be made using non - parametric methods such as the Wilcoxon signed - rank test.

We will incorporate these diagnostic and goodness - of - fit techniques into our statistical analysis plan. Before conducting the main repeated measures analyses, we will first perform the normality tests. Based on the results, we will decide whether to proceed with the standard repeated measures ANOVA or use one of the alternative methods for non - normal data.

Reviewer #2:

1.This article provides a new acupuncture treatment approach for early to mid-stage depression. I have two suggestions for the authors' consideration.

Could you please attach a reference picture of the sham acupuncture? This would allow readers to more intuitively understand how the sham acupuncture procedure was performed.

ANSWER:

Thank you for your valuable suggestion to enhance the clarity of our sham acupuncture procedure. We appreciate your emphasis on providing intuitive visual aids for readers. Following your recommendation, we have now included a detailed schematic diagram of the sham acupoint locations in Supplementary Material S3. This figure illustrates the precise positioning of the sham acupoints (located 1 inch lateral to the true acupoints, in non-meridian, non-acupoint regions) relative to the anatomical landmarks and true acupoints.

2.Previous clinical or experimental studies on acupuncture for depression exist. Please discuss in the manuscript how the Jin's Three Needle technique differs from and what advantages it may have over other acupuncture approaches studied.

ANSWER:

Thank you for your constructive suggestion to elaborate on the distinctiveness and potential advantages of the Jin’s Three-Needle (JTN) technique compared to other acupuncture approaches f

---

## [Decision Letter · Decision Letter 1]

21 Nov 2025

Acupuncture (Jin's Three-Needle) versus sham acupuncture in Treating Mild-to-Moderate Depression: Study Protocol for Randomized Clinical Trial

PONE-D-25-44858R1

Dear Dr. Wen,

We’re pleased to inform you that your manuscript has been judged scientifically suitable for publication and will be formally accepted for publication once it meets all outstanding technical requirements.

Kind regards,

Yung-Hsiang Chen, Ph.D.

Academic Editor

PLOS ONE

Additional Editor Comments (optional):

Congratulations on the acceptance of your manuscript, and thank you for your interest in submitting your work to PLOS ONE.

Reviewers' comments:

Reviewer's Responses to Questions

**Comments to the Author**

1. Does the manuscript provide a valid rationale for the proposed study, with clearly identified and justified research questions?

Reviewer #1: Yes

2. Is the protocol technically sound and planned in a manner that will lead to a meaningful outcome and allow testing the stated hypotheses?

Reviewer #1: Yes

3. Is the methodology feasible and described in sufficient detail to allow the work to be replicable?

Reviewer #1: Yes

4. Have the authors described where all data underlying the findings will be made available when the study is complete?

Reviewer #1: Yes

5. Is the manuscript presented in an intelligible fashion and written in standard English?

Reviewer #1: Yes

You may also provide optional suggestions and comments to authors that they might find helpful in planning their study.

Reviewer #1: I don't agree with some of the answers, but they are well-thought out. I have a lot of problems blinding the study statistician, who is the one person in the study who needs to know the treatment groups. If your statistician cannot be trusted to be unbiased, that is problematic.

**Do you want your identity to be public for this peer review?** For information about this choice, including consent withdrawal, please see our Privacy Policy

Reviewer #1: No

---

## [Editor Report · Acceptance letter]

PONE-D-25-44858R1

PLOS ONE

Dear Dr. Wen,

I'm pleased to inform you that your manuscript has been deemed suitable for publication in PLOS ONE. Congratulations! Your manuscript is now being handed over to our production team.

Kind regards,

on behalf of

Dr. Yung-Hsiang Chen

Academic Editor

PLOS ONE